# END-EFFECTOR-ELBOW: A NEW ACTION SPACE FOR ROBOT LEARNING

## ABSTRACT

Joint and end-effector are the two most dominant action spaces for robot arms within the robot learning literature. Joint actions, while precise, often suffer from inefficient training; end-effector actions boast data-efficient training but sacrifice the ability to perform tasks in confined spaces due to limited control over the robot joint configuration. This paper introduces a novel action space formulation: End-Effector-Elbow (E3), which addresses the limitations of existing paradigms by allowing the control of both the end-effector and elbow of the robot. E3 combines the advantages of both joint and end-effector action spaces, offering fine-grained comprehensive control with overactuated robot arms whilst achieving highly efficient robot learning. E3 systematically outperforms other action spaces, when precise control over the robot configuration is required, both in simulated and real environments.
**Project website:**  https://doubleblind-repos.github.io/

## 1 INTRODUCTION

In robot learning, the choice of the action representation greatly affects task learning efficiency and precision of robot control, as it determines the action space containing all the agent's learnable actions. There are two predominant action spaces in the current landscape of robot learning research: joint coordinates and end-effector coordinates. Joint coordinates provide joint-level control over the entire configuration of the robot but typically is less sample efficient when training. End-effector coordinates simplify the learning process by aligning action and task space but cannot provide full-body control in overactuated arm configurations. For example, as shown in Figure 2.b, it is hard for the robot to stretch deep into a cabinet to grasp a cup due to the free-motion in the elbow.

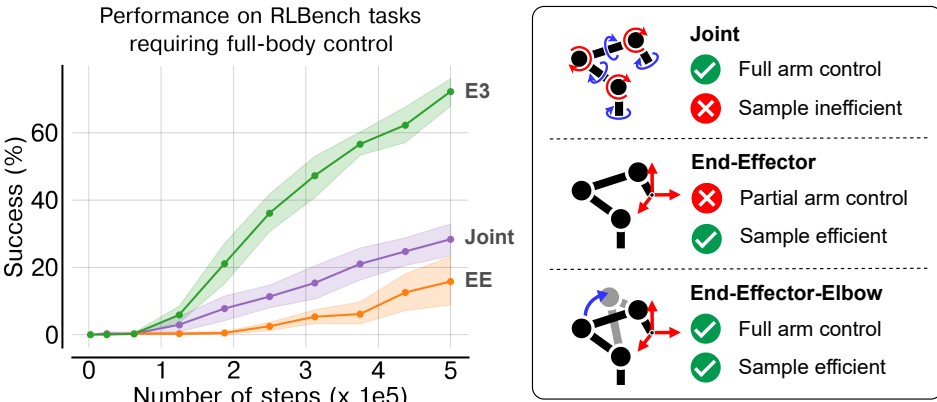

Figure 1: The End-Effector-Elbow (E3) action space combines learning efficiency (seen in End-effector space) and full control over the arm configuration (seen in joint space). E3 strongly outperforms other action spaces in tasks where control over the entire robot configuration is required or helpful.

To address the shortcomings of these two action spaces, we propose a novel action space formulation for overactuated arms: **E**nd-**E**ffector **E**lbow (**E3**). The E3 action space enhances the existing EE action space by introducing additional control over the elbow position. This improvement can enable both efficient learning and full control over the morphology of the robot. The E3 action space can be parameterised and controlled in various ways, and in this paper, we explore two specific implementations: **E3A**ngle (**E3A**) and **E3J**oint (**E3J**). E3A allows direct control of the elbow by controlling its angle, while E3J allows indirect control of the elbow by controlling one of the joints separately to influence the elbow position. We found that, while both methods allow precise elbow control, the E3J action space seems more suited for robot learning, leading to higher performance, and so we bring it forward as our main E3 parameterisation for future applications.

Across a series of quantitative RLBench (James et al., 2020) reinforcement learning (RL) experiments, we demonstrate that E3 significantly outperforms both joint and EE action spaces while also being more sample efficient. Moreover, we show that the benefits of this new action space extend to real-world settings via a set of qualitative real-world imitation learning (IL) experiments.

**Contributions.** Our contributions can be summarised as follows:

- we present a new action space: End-Effector-Elbow (E3), along with two concrete realisations of it: E3Angle (E3A) and E3Joint (E3J). We empirically validate them, showing that both enable precise control, with E3J (our main proposition) showing sample efficient learning through the alignment of task and action space;

- an extensive RL-based evaluation of our approach in simulation with RLBench (James et al., 2020; Rohmer et al., 2013; James et al., 2019), showing that by simply swapping the action space, our approach becomes extremely relevant in tasks requiring precise control over the arm configuration, without compromising performance in other tasks;

- a real-world IL application of our approach that demonstrates the applicability of the approach, succeeding in tasks where the EE action space fails.

## 2 RELATED WORK

**Action spaces for robotic manipulation.** When controlling a robot arm, commands need to be sent to the controller in configuration space, e.g. joint positions or velocities. However, it is possible to work in Cartesian task space coordinates and then translate these into configurations using an inverse kinematics (IK) algorithm, e.g. EE, and operational space controllers (Nakanishi et al., 2008). Previous work tends to associate working in task space with higher performance (Matas et al., 2018; Varin et al., 2019; Martín-Martín et al., 2019; Zhu et al., 2020). However, the tasks analysed generally only require partial control over the robot joint configuration. Recent work has attempted to link joint control to task space coordinates, either learning a joint space controller that can drive the agent in task space coordinates (Kumar et al., 2021b) or using the forward kinematics to translate joint actions into EE actions and learning multi-action space policies (Ganapathi et al., 2022). While joint and EE controllers are both common for low-level fine-grained control, other work has focused on the problem of solving long-term tasks or low data efficiency adopting high-level action primitives for control, such as pushing and grasping (Zeng et al., 2018), sliding and turning (Dalal et al., 2021; Nasiriany et al., 2022), 3D visual coordinates in the workspace (James & Davison, 2022), or a hierarchical combination of task-space controllers (Sharma et al., 2020).

**Robot learning.** Learning paradigms for robotics, such as reinforcement learning (Fujimoto et al., 2018; Haarnoja et al., 2018) and imitation learning (Pomerleau, 1988; Ho & Ermon, 2016; Florence et al., 2022), or a combination of both (Rajeswaran et al., 2017; Zhan et al., 2021), can help automate behaviour learning in robotics. While previous work has shown the potential of robot learning when training from large chunks of data (Kalashnikov et al., 2018; Akkaya et al., 2019; Lu et al., 2021), recent work has proven that robot learning can lead to data-efficient learning too (Smith et al., 2022; Zhao et al., 2023). There are several challenges that are specific to robot learning, such as extracting useful features from high-dimensional observations (Radosavovic et al., 2023; Nair et al., 2022), choosing adequate policy parameterizations (Seyde et al., 2021; James & Abbeel, 2022a), and online adaptation (Kumar et al., 2021a). In this work, we focus on the problem of adopting the optimal action space, in order to allow both precise control and efficient learning.

**Redundant manipulators control.** Despite 6 DoF robot arms being sufficient for many tasks and industrial settings, the redundancy in 7 DoF can be useful for manipulability (Yoshikawa, 1985a;b), torque optimisation (Suh & Hollerbach, 1987), obstacle avoidance (Nakamura et al., 1987) and singularity robustness (Chiaverini, 1997; Cheng et al., 1998). Redundancy resolution is a long-standing problem in robotics (Hollerbach & Suh, 1987; Hsu et al., 1989; Seraji, 1989) whose solutions often require taking into account the structure of the robot, by defining additional constraints (Howard et al., 2009; Lin et al., 2015), e.g. an arm angle (Shimizu et al., 2008), and/or analytical solutions (He & Liu, 2021), or defining constraints in the robot's joints, as done in (Ratliff et al., 2018), where control over one one joint is combined with other task-space controllers, using Riemannian motion policies.. To the best of our knowledge, this work represents the first to study how varying the action space for parameterised control of redundancies affects robot learning performance for overactuated arms.

## 3 ACTION SPACES FOR ROBOT LEARNING

In this section, we explore the two predominant robot arm control methods within the current landscape of robot learning research: end-effector and joint coordinates.

### 3.1 END-EFFECTOR ACTIONS

**Definition 3.1 (End-Effector Action Space)** *The End-Effector (EE) action space considers an action space $\mathcal{A}_{ee}$ with 6 DoF, consisting of $\mathcal{A}_T \in \mathbb{R}^3$ for translations, and an over-parameterized quaternion action $\mathcal{A}_R \in \mathbb{R}^4$ for rotations of the end-effector.*

The End-Effector action space inherently conforms to the task space, wherein the objective is to manipulate the end-effector of the robot for environmental interaction. This alignment has been shown to improve sample efficiency by simplifying the optimisation process (Matas et al., 2018; Plappert et al., 2018). However, it falls short of providing comprehensive control for overactuated robot arms. For example, the redundancy of overactuation arises when controlling a 7 DoF manipulation with the 6 DoF EE action space, where multiple joint configurations may achieve the same desired end-effector pose. As a result, the joint space remains under controlled and the "elbow" joint can vary whilst maintaining the EE pose. This free-motion is described by the intersection of two spheres seen in Figure 2.a. This limitation becomes evident in scenarios where precise control of the elbow position of the robot is imperative for successful task execution, such as obstacle avoidance or reaching into confined spaces like cramped cupboards, as shown in Figure 2.b. In these situations, EE actions may inadvertently lead to collisions and sub-optimal outcomes.

### 3.2 JOINT ACTIONS

**Definition 3.2 (Joint Action Space)** *The Joint action space considers an action space $\mathcal{A}_J \in \mathbb{R}^n$ where $n$ is the number of joints which make up the robot arm.*

On the other hand, joint actions are adept at addressing collision avoidance concerns through complete joint-level control of the robot. With access to all the joints of the robot, controlling the joints allows the robot to accurately reach any configuration and address the issue discussed in Figure 2.b. However, $\mathcal{A}_J$ is not aligned with the Cartesian task space the robot is interacting with. Mapping from $\mathcal{A}_J$ to this task space requires the policy to understand the underlying non-linear kinematics of the robot, which, nevertheless, introduces a more complex learning process to the agent. Consequently, the joint action space often has worse sample efficiency than EE action space.

## 4 END-EFFECTOR-ELBOW ACTION SPACE FOR ROBOT LEARNING

In this work, we introduce End-Effector-Elbow (E3) action spaces, a new family of action spaces for robot learning with overactuacted arms. E3 action spaces combine the best of both EE and joint actions, enabling accurate control of the entire arm and allowing the agent to act in task space. The section is organised as follows. We first give formal definitions to the pivotal criteria of action spaces for sample efficient and generalisable robot learning, followed by formal definition to the E3 action space and its two realisations.

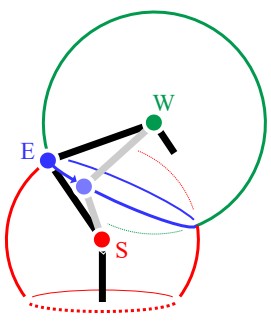
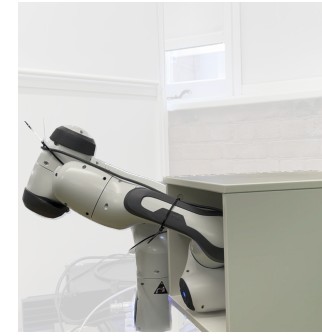
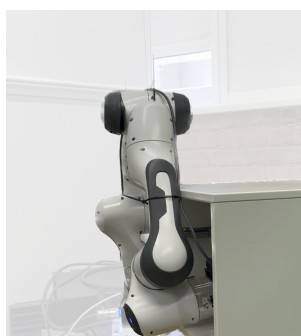

(a) Free-motion of the elbow

(b) Manipulation in confined space

Figure 2: **(a)** The position of the arm elbow (E) depends on both the shoulder (S) and the wrist (W). Given fixed W and S, E can move freely around the blue circle whilst maintaining EE pose. This circle is the intersection of the red sphere determined by S and the green sphere determined by W. Illustrations are shown in the Appendix A.1 which show the free-motion of the elbow on a range of consumer overactuated robots. **(b)** The position of the elbow is crucial for manipulation in confined spaces, e.g., removing an object from the cabinet. A correct elbow position enables the robot to enter the cabinet (left), but an incorrect elbow position causes unsuccessful execution of the action (right).

### 4.1 PIVOTAL CRITERIA OF ACTION SPACES FOR ROBOT MANIPULATION

The trade-off between controllability and flexibility leads to the question: does an alternative action space which strikes a balance between these aspects exist?

Before we discuss the exact realisation of the alternative action mode, we first define three pivotal criteria of action spaces for robot manipulation, in lieu of the aforementioned precision (comprehensive control of the robot), efficiency (task learning without extensive training periods) and flexibility (allowing the robot to perform tasks with varying degrees of complexity and constraints).

- *Validity*: the policy outputs of the agent need to yield achievable configurations given the kinematics constraints and joint limits of the robot. While agents can learn to avoid invalid actions through applying punitive rewards, a primary objective of the action space should revolve around reducing the likelihood of such occurrences;

- *Consistency*: we expect the robot configuration to be consistent for identical state-action pairs, i.e. no free-motion. As part of this consistency criterion, post-processing of the actions, such as clipping or filtering, should also be minimised;

- *Alignment*: the alignment between the action space and the task space is beneficial for sample efficiency. In the context of robotic manipulation, the task space is conventionally defined by 6 DoF which describe the position and orientation of the gripper. The pose of any rigid objects in a task can also be described with these six variables; thus, it is straightforward to describe grasp locations or movement trajectories.

The EE action space naturally satisfies the alignment criteria since its action space $\mathcal{A}_{ee}$ is consistent with the task space, but due to the redundancy of the 7 DoF robot arm, it lacks validity and consistency as certain poses might violate the kinematics and environment constraints. The joint action space, on the other hand, provides both validity and consistency, but fails to meet the criteria for alignment.

### 4.2 END-EFFECTOR-ELBOW ACTION SPACE

In this section, we introduce End-Effector-Elbow (E3): an action space for robot learning with overactuated arms that aims to meet all validity, consistency, and alignment criteria outlined above.

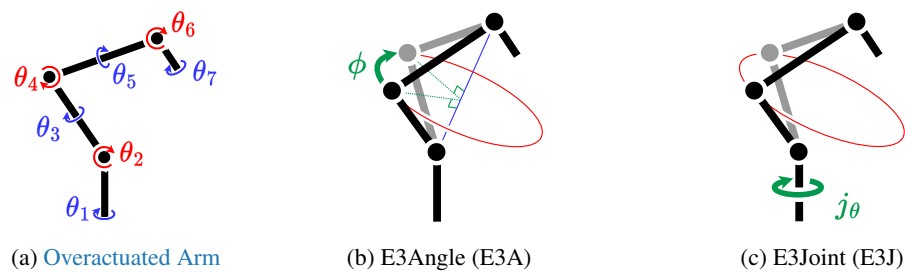

(a) Overactuated Arm        (b) E3Angle (E3A)        (c) E3Joint (E3J)

Figure 3: **End-Effector-Elbow (E3) family of action spaces.** The elbow free-motion circle is shown in red. (a) collinear joints are shown in blue and orthogonal joints in red (b) E3A is constrained directly by the angle of the elbow around the axis from the shoulder to the wrist ($\phi$). (c) E3J is constrained indirectly through removing a joint (in this case the base joint, $j_\theta$) from the inverse kinematics, and controlling this joint separately.

**Definition 4.1 (E3 Action Space)** *E3 Action space defines an action space of $\mathcal{A}_{e3} = (\mathcal{A}_{ee}, \mathcal{A}_{elbow})$, which constrains the free-motion present in overactuated arms when using end-effector pose control.*

Given a $N$ DoF arm (where $N \geq 7$), E3 action space builds upon the EE action space, adding $N-6$ dimensions to the action which constrain the free motion of the elbow, as shown in Figure 2a with a 7 DoF manipulator. Thus, by maintaining Cartesian control over the end-effector and parameterisation redundancies, the E3 action space aims to combine the sample efficiency of EE action space with full control over the joint configuration.

By adopting an appropriate parameterisation to address the redundancy control problems, we can control these additional free-motions which are unavailable to EE action space. In practice, we visually identify the redundancy into the possibility of controlling the elbow, which derives the name of our method: End-Effector-Elbow (E3). While the E3 action space represents a general formulation, we concentrate our attention on 7-degree-of-freedom (DoF) robot arms — a common choice of manipulator in recent literature. Two different realisations of the E3 action space will be introduced in the rest of this section.

### 4.3 E3ANGLE - ARM ANGLE CONTROL

In classical control literature, one popular solution to address the redundancy control problem, is to describe the redundancy through the adoption of an additional angle, often referred to as the arm angle (Shimizu et al., 2008). For a 7 DoF robot arm, we first identify the free-motion circle, defined as the intersection of two spheres centred at the adjacent joints (as shown in Figure 2a) and define the **E**nd-**E**ffector-**E**lbow **Angle** (**E3Angle** or **E3A**) as:

**Definition 4.2 (E3A)** *End-Effector-Elbow action space introduces an action space $\mathcal{A}_{E3A} = (\mathcal{A}_{ee}, \phi)$, where $\phi$ is the rotation angle around the line connecting the centre of two spheres (see Figure 3b).*

With a known robot kinematics model, we show for a given gripper pose and rotation angle $\phi$, there exists a closed-form solution to the position of the elbow, $\mathbf{p_E} \in \mathbb{R}^3$

$$\mathbf{p_E} = \mathbf{c_E} + r_E \cdot (\mathbf{t_E} \cdot \cos(\phi) + \mathbf{b_E} \cdot \sin(\phi)) \tag{1}$$

where $c_E \in \mathbb{R}^3$ and $r_E$ are the centre and radius of the elbow free-motion circle, and $b_E \in \mathbb{R}^3$ and $t_E \in \mathbb{R}^3$ are bi-tangent the tangent to this circle at the desired elbow angle $\phi$ (where $0°$ is the highest point on the circle). The formal derivations are deferred to the Appendix (see A.2). We can now add $\mathbf{p_E}$ as an additional constraint to the IK solver and obtain the joint positions of the arm.

E3A successfully addresses the joint redundancy issue of a 7 DoF arm and satisfies the consistency and alignment criteria. Nevertheless, with the additional IK constraint imposed on the elbow joint, the solution of the IK problem will face a more complex optimisation landscape. As a result, we empirically observe that, using E3A, the agent struggles to find valid IK solutions, in some cases leading to worse training performance.

### 4.4 E3JOINT - JOINT-FIRST-CONTROL

To address the invalid IK solution issue of E3A, we further introduce **E**nd-**E**ffector-**E**lbow **Joint** (**E3Joint** or **E3J**), which removes the additional constraint by directly controlling one of the joint positions of a robot arm. Formally, we define E3J as

**Definition 4.3 (E3J)** *End-Effector-Elbow Joint action space has an action space $\mathcal{A} = (\mathcal{A}_{ee}, j_\theta)$, where $j_\theta \in \mathbb{R}$ is a scalar action that controls the position of a chosen arm joint.*

Controlling an additional joint position can be seen as a way of reducing the number of optimising variables solved with the IK. Specifically, we set the chosen joint to the desired position and control the remaining $7 - 1 = 6$ DoF of the robot in the task space by solving the IK. Ideally, E3J will allow the arm to obtain both the flexible control to the robot, and a high IK solver success rate, given that less constraints are imposed. The E3J method would also easily scale to robot arms with higher DoFs than 7, where it requires controlling N - 6 joints separately so that the 6 DoFs of the end-effector (position and orientation) are constrained by the task space coordinates, while the remaining joints control the redundant dimensions.

**Joint selection.** The selection of the joints that are controlled separately is crucial, as this influences how the IK problem is solved for the other joints. Consider the forward kinematics of a 7 DoF robot as a chain (see Figure 3a). Intuitively, fixing the intermediate joints will break the chain. Although this does not introduce additional constraints to the optimisation problem of solving the IK, extra efforts are needed to customise IK solvers as the chaining property of the robot kinematics is broken. Thus, the first (base) or the last joint (wrist) are more practical choices. These two joints affect the robot configuration in different ways: by rotating the first joint (which controls the shoulder rotation) the entire robot configuration rotates by the same angle, changing the end effector position and its orientation with respect to the $z$ axis; by rotating the last joint (which controls the wrist rotation) the end effector orientation is rotated around the wrist axis accordingly.

## 5 EXPERIMENTS

In our experiments, we aim to show the usefulness of the E3 action space in several tasks, compared to the common EE and joint action modes. We also aim to analyse how different implementations of E3 impact robot learning performance. For this purpose, we present results both in simulation (with delta action spaces) and real-world settings (with absolute action spaces), using RL and IL, respectively. Both in simulation and in the real world, we adopt the Franka Emika 7 DoF arm for our experiments, nonetheless, we stress that our action space can be easily adapted to any 7 DoF arm, especially E3J as it requires no additional analysis of the robot kinematics. We also highlight that, in both settings, we employ IK-based controllers, whose details are provided in Appendix.

### 5.1 REINFORCEMENT LEARNING IN SIMULATION

We train RL agents in RLBench (James et al., 2020), across two sets of tasks: (i) 6 '**Full-body tasks**' tasks which specifically require or benefit from full control of the robot joint configuration, and (ii) 8 '**Vanilla RLBench**' tasks that are commonly employed for RL evaluation. In both settings, we adopt SAC (Haarnoja et al., 2018), and provide dense reward functions for all tasks, to drive the agent's exploration towards success. Hyperparameters and the reward engineering process can be found in Appendix.

**Full-body tasks.** Building on top of RLBench (James et al., 2020), we evaluate our approach on a set of 6 tasks where full control of the robot joint configuration is required to carry out the task successfully and consistently. Among these tasks, 4 tasks are completely new, and two tasks ('meat off grill' and 'take cup out cabinet') are from the original RLBench tasks. Tasks are described in Appendix A.3, where we explain why full body control is important.

Overall final performance after 500k steps across the environments, is presented in Figure 1 (E3 is represented by E3J). Detailed performance over time for each task is presented in Figure 4. We observe that the overall performance obtained using E3J is much higher than with EE and joint. In particular, looking at the tasks in detail, we see that in tasks which are simpler but require correct elbow movement (first row), joint actions still allows to solve the tasks, despite being less efficient

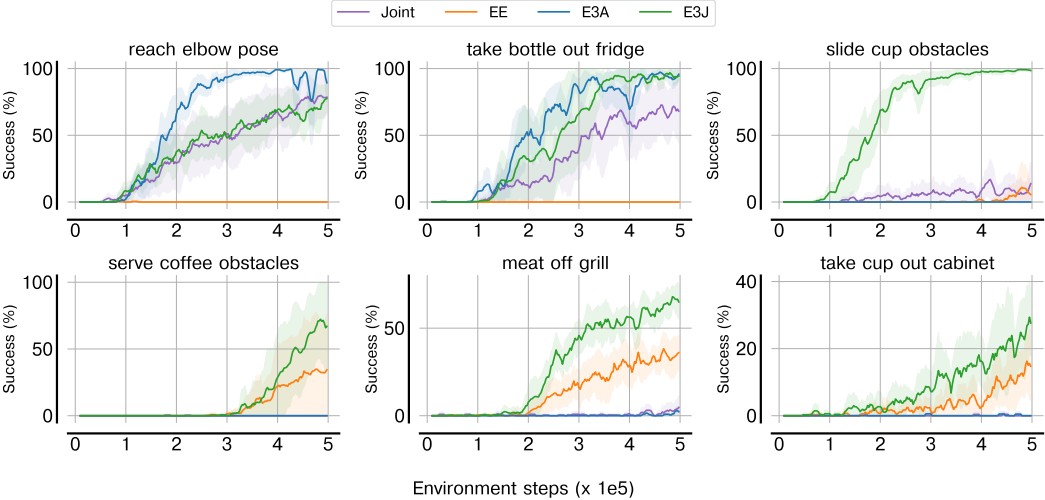

Figure 4: **Full-body tasks results.** Results on 6 tasks where control over the entire robot arm configuration is required to succeed or perform better. Videos available on the project website.

than E3J. As we move to more complex tasks, where control of the elbow is not required to attain task success, but provides efficient obstacle avoidance, EE is able to solve the tasks, but both EE and joint actions systematically perform worse than E3J. E3A works well in simple tasks that require accurate configuration control, outperforming all other approaches, including E3J, but loses in data efficiency when compared to E3J and EE in the other, more complex, tasks.

**Vanilla RLBench tasks.** We study the performance of the E3 action space in a set of standard RLBench tasks that have previously been used in several works (James & Davison, 2022; Liu et al., 2022; James et al., 2022; Zhao et al., 2022; James & Abbeel, 2022c;b; Adeniji et al., 2023; Seo et al., 2023). These tasks do not require elbow positioning as there are no obstacles to avoid or precise poses to achieve. Thus, the objective of evaluating these is to show that the learning efficiency coming from working in task space, as EE does, can be preserved with E3 methods.

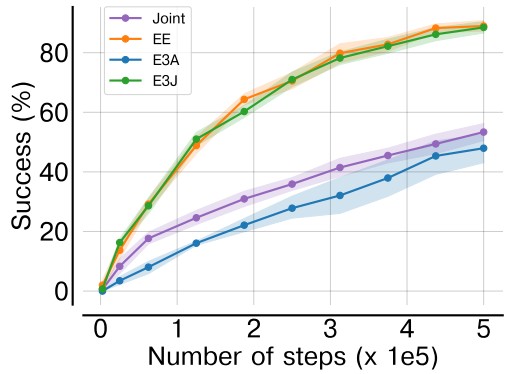

Figure 5: **Performance on RLBench tasks.** EE and E3J share the best performance as expected, as they have similar alignment and validity. E3A also preserves alignment but makes it difficult to provide valid actions.

In Figure 5, we present performance over time, aggregated with stratified bootstrapped sampling with 50k repetitions, using RLiable (Agarwal et al., 2021). Detailed training curves per task are presented in the Appendix. It can be observed that the performance of EE and E3J is completely on par, both in terms of final performance and learning efficiency. Using the joint action space, which is not aligned with the space, leads to slower learning. E3A also learns slower and we hypothesise this is due to the complexity of predicting valid {EE,arm angle} configurations. Further insights are provided in the remainder of this section.

**Analysis.** As discussed in Section 4, the E3 action space is prone to multiple implementations. We presented two, E3A and E3J, proposing to use the latter, with control over the base shoulder rotation joint (first joint), as the main method. Here, we provide empirical justification for our choice through two ablations. In Figure 6a we compare the aggregated performance over time of different implementations of our method on the 6 elbow tasks: E3J-BASE, controlling the base joint separately, E3J-WRIST, controlling the wrist joint instead, and E3A. For completeness, we also plot

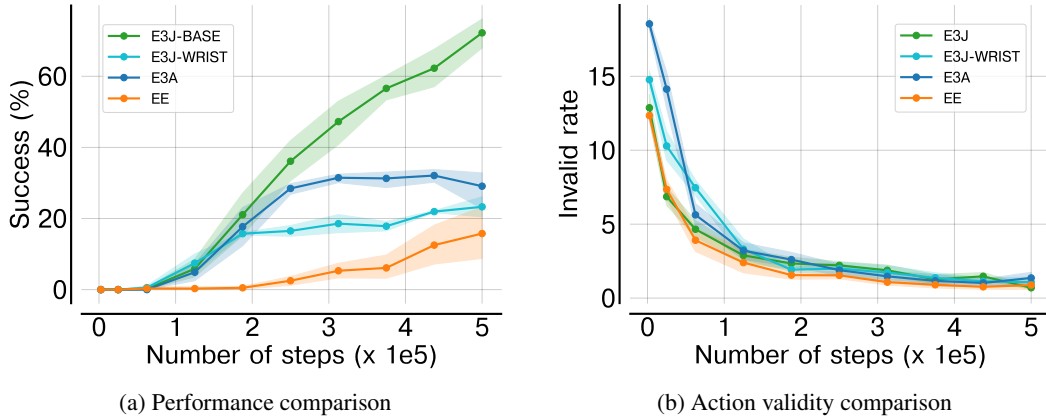

(a) Performance comparison          (b) Action validity comparison

Figure 6: **Ablation study.** Comparing performance and validity of actions for different E3 reali-sations on the full-body tasks. For both plots, the data is aggregated with a stratified bootstrapped sampling strategy with 50k repetitions, using RLiable (Agarwal et al., 2021).

EE results. We observe that while all E3 methods perform better than plain EE, our main choice (E3J-BASE) attains much higher performance than the other implementations.

To provide insights into the potential reasons for our E3J implementation being the most suitable for robot learning, we can look into the validity of the actions provided by the agent with the different methods. Note that invalid actions can be caused by the Jacobian-based IK solver failing due to an impossible configuration request. This could be due to: (i) joint configurations that are too far from the current one and thus unreachable, (ii) attempts to collide with objects, or (iii) configurations outside the joint limits. As OSC controllers (Khatib, 1987) are also commonly adopted in RL, when using delta action spaces, in the future, it would be worth studying the effects of adopting such kinds of controllers with the different flavours of E3, and compare them with the IK-based results, in terms of performance and validity.

In Figure 6b, we show how the rate of invalid actions decreases over time for all methods. However, there is a major gap between our E3J choice (i.e. E3J-BASE), E3J-WRIST, and E3A, especially in the initial learning phase. The agent struggles more to find valid actions for E3J-WRIST and E3A, potentially leading to (over) cautious behaviours later, hindering performance of the agent. Instead, the E3J-BASE invalid actions rate is in line with the EE rate, showing how this implementation provides greater flexibility and performance, without hindering task learning.

## 5.2 IMITATION LEARNING IN REAL WORLD

The use of E3 is not limited to reinforcement learning; in this section we complete a **qualitative** exploration into its use as an action space for real world imitation learning (IL). In Figure 7, we show pictures of a real-world adaptation of the 'take cup out cabinet' task we adopted in simulation. The cabinet has a simpler structure than that in simulation, but it is twice as deep (the real cabinet being $> 0.3$ m deep); additionally, the narrow stem of the cup requires precise grasping in comparison with RLBench grasp mechanics.

Due to the confined workspace of these tasks, solving these problems with IL requires the action space to reproduce the full robot configuration seen in the demos. It is therefore expected that both E3 and Joint should be able to complete the task, while EE success is not guaranteed.

We collect 29 tele-op demonstrations (23 for training and 6 for validation), which we divide into two subtasks: (i) '**Reach Cup**': starting from the top of the cabinet, reaching the cup inside the cabinet and grasping it; (ii) '**Remove Cup**': starting from inside the cabinet, grasping and taking the cup out of the cabinet. Both subtasks require precise end-effector pose control, to reach and grasp the cup, and full control over of robot joint configuration, to consistently avoid colliding with the cabinet. 'Reach Cup' requires a correct change in elbow angle to achieve success whilst 'Remove Cup' can be achieved without changing the elbow angle (since the subtask is initialised inside the

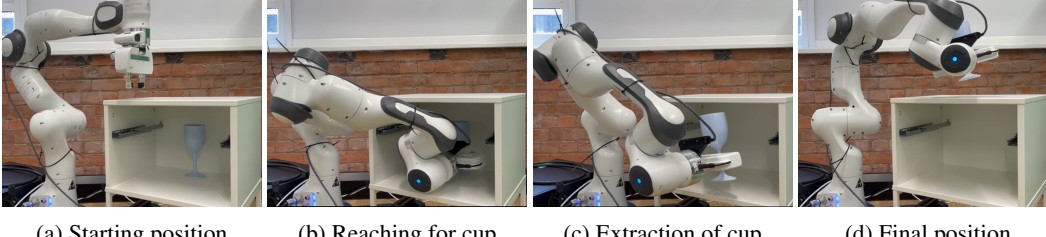

| (a) Starting position | (b) Reaching for cup | (c) Extraction of cup | (d) Final position |

Figure 7: **Real world qualitative experiment.** We learn two tasks with imitation learning: '**Reach Cup**', which involves reaching the cup inside the cabinet and grasping it, and '**Remove Cup**', which involves grasping and taking the cup out of the cabinet. Results are best seen through videos of the inference, found on the project website.

cupboard). For training the agent we use the same strategy as in ACT (Zhao et al., 2023), employing action chunking and temporal ensembling over multi-step predictions. Training details and model architecture are presented in Appendix.

Results across all the action spaces are shown in Table 1, where each task was run 10 times per action space. As expected, we observe that both joint and E3J successfully solve the task along with both subtasks. However, EE is not able to enter the cabinet and can (sometimes) extract the cup only when starting inside the cabinet. Apart from EE colliding with the cabinet upon entry, all other failures where caused by inaccurate grasping.

Table 1: **Real world results.** Comparing success rate of different action spaces. As expected, E3J and Joint share the best performance, as they both have full control over the robot configuration.

|       | Reach Cup | Remove Cup | Full task |
|-------|-----------|------------|-----------|
| Joint | 100%      | 90%        | 90%       |
| EE    | 0%        | 40%        | 0%        |
| E3J   | 100%      | 80%        | 90%       |

## 6 CONCLUSION

We presented E3, a new action space formulation that allows to achieve both precise control and efficient robot learning, for overactuated arms, overcoming the shortcomings of the previous standard joint and end-effector action spaces. By putting emphasis on the issues which arise when not controlling the elbow configuration, we show the importance of switching to new action representations as robot learning advances and moves on to more complex tasks, requiring precision and the ability to avoid obstacles. Our E3J solution demonstrates ease of implementation and has exhibited consistent success in diverse settings, spanning both simulation and real-world scenarios. This suggests its potential readiness to universally supplant both EE and joint methods in the field of robot learning.

In this work, we focussed on low-level fine-grained control, where the agent directly outputs task-space coordinates and/or joint coordinates. There are several opportunities for extending the idea of E3 to more complex settings. For instance, we could exploit the idea behind E3A, of parameterizing the redundancy with an explicit arm angle, to develop high-level primitives such as 'turn the elbow', to integrate in a more hierarchical RL controller (Dalal et al., 2021; Nasiriany et al., 2022). Another opportunity arises from the idea of flexibly composing controllers that satisfy task-space constraints Sharma et al. (2020); Ratliff et al. (2018). By combining this approach with the idea of flexibly controlling redundant joint positions, as in E3J, we could build a more general framework that allows selecting which joints to control depending on the task, rather than having to select one beforehand.

**Reproducibility Statement** In order to ensure reproducibility of our results:

- we provide details on the E3Angle method realisation in the Appendix;
- we share the code for the simulation experiments;
- we specify the number of seeds for all experiments in the Appendix;
- we used RLiable (Agarwal et al., 2021) for Figures 2, 5, and 6 for providing statistically robust results.

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

# A   APPENDIX

## A.1   EXAMPLES OF ARM ANGLES IN REAL ROBOT MODELS

Showing the arm angle on real robot models (rendered in CoppeliaSim).

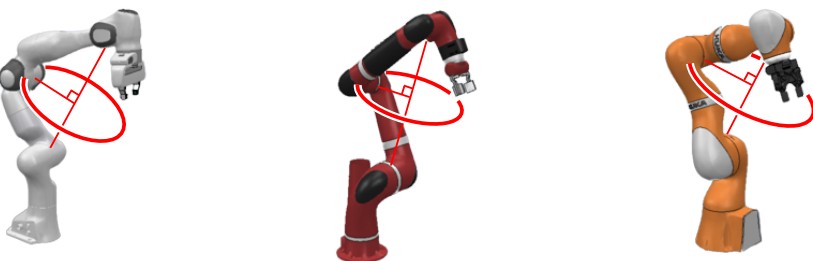

Figure 8: Showing the arm angle emerging from the redundancy of these 7 DoF robot arms. From left to right: Franka Emika Panda, Sawyer, and KUKA LBR iiwa 14 R820.

## A.2   E3ANGLE DETAILS

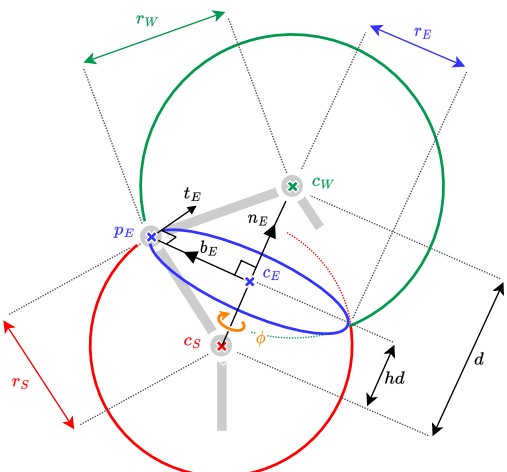

Figure 9: **Calculating position of elbow.** The position of the elbow for a given $\phi$ can be calculated by finding the intersection of two spheres.

Given we know $r_S$, $r_W$, $\mathbf{c_S} \in \mathbb{R}^3$, and $\mathbf{c_W} \in \mathbb{R}^3$ we can find the equation of the elbow free-motion circle (parameterised by $\phi$), as shown in Fig 9; and thus, the position of the elbow ($\mathbf{p_E} \in \mathbb{R}^3$) for a given angle ($\phi$).

$$d = \|\mathbf{c_W} - \mathbf{c_S}\|$$

$$\mathbf{n_E} = \frac{\mathbf{c_W} - \mathbf{c_S}}{d}$$

$$\mathbf{t_E} = \begin{cases} \frac{\mathbf{t}}{\|\mathbf{t}\|}, & \text{if } \mathbf{n_E} = \mathbf{k} \text{ where } \mathbf{t} = \mathbf{n_E} \times \mathbf{k} \\ -\mathbf{j}, & \text{otherwise} \end{cases}$$

$$\mathbf{b_E} = \begin{cases} \frac{\mathbf{b}}{\|\mathbf{b}\|}, & \text{if } \mathbf{n_E} = \mathbf{j} \text{ where } \mathbf{b} = \mathbf{n_E} \times \mathbf{j} \\ -\mathbf{i}, & \text{otherwise} \end{cases}$$

If $d = r_S + r_W$ (there is no solution if $d > r_S + r_W$):

$$\mathbf{p_E} = \mathbf{c_S} + \frac{r_S}{d} \cdot (\mathbf{c_W} - \mathbf{c_S})$$

Else:

$$h = \frac{1}{2} + \frac{r_S^2 - r_W^2}{2d^2}$$

$$\mathbf{c_E} = \mathbf{c_S} + h \cdot (\mathbf{c_W} - \mathbf{c_S})$$

$$r_E = \sqrt{r_S^2 - (h \cdot d)^2}$$

$$\mathbf{p_E} = \mathbf{c_E} + r_E \cdot (\mathbf{t_E} \cdot \cos(\phi) + \mathbf{b_E} \cdot \sin(\phi))$$

### A.3 TASK DETAILS

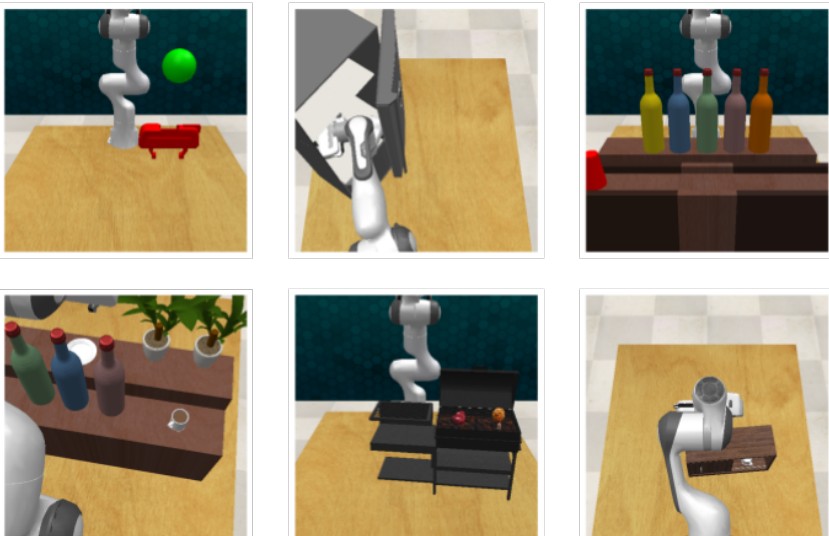

Figure 10: **Visualization of the full-body tasks.** *(Top row)* reach elbow pose, take bottle out fridge, slide cup cabinet. *(Bottom row)* serve coffee obstacles, meat off grill, take cup out cabinet.

**Full-body tasks.** Building on top of RLBench (James et al., 2020), we evaluate our approach on a set of 6 tasks where full control of the robot joint configuration is important in order to carry out the task successfully. Among these tasks, 4 tasks are completely new, and two tasks ('meat off grill' and 'take cup out cabinet') are from the original RLBench tasks. Here we describe the task briefly, explaining why control over the elbow is important in these tasks:

- *reach elbow pose*: the agent is asked to reach a specific configuration, by visually imitating a fake red gripper in the environment and positioning the elbow in a green sphere. Fake gripper and sphere are floating and their positions are randomised;
- *bottle out fridge*: the agent starts with the gripper inside the fridge, holding a bottle. The agent is required to open the fridge and takes the bottle out, without releasing the bottle or letting it hit the inside of the fridge.
- *slide cup obstacles*: the agent is asked to slide a cup across a table, with obstacles to avoid in order to perform the sliding action.

- *serve coffee obstacles*: the agent needs to serve a cup of coffee on a plate, in a cluttered scene, while keeping the orientation of the cup vertical and not letting the cup hit any objects.
- *meat off grill*: the agent needs to take a chicken thigh off the grill and place it next to it, in the cooling area. The grill is open, with the lid constituting an obstacle, and its position is randomised.
- *take cup out cabinet*: the agent is asked to take a cup out of a cabinet. The cabinet can be entered only from side and its position is randomised. To make the task easier, we start with the cabinet open, compared to the original task.

**Reward engineering.** To provide dense reward functions, we adopt a common interface for designing rewards for all tasks. We analyse the success conditions and design a subset of operations that should be executed accordingly. For instance, the 'stack_wine' task requires the wine to be placed on the rack: we divide the task into: i) grasping the wine, ii) placing it on the rack. Each subset operation has a reward varying between 0 and 1, where 1 means that the operation has been completed.

We try to leave the agent as free as possible to choose any path for completing the task, unless we found it was necessary to drive the agent to a specific position for completing the task, e.g. the 'put rubbish in bin' task requires the agent to first be driven on top of the rubbish basket, before rewarding the agent for placing the object inside it, otherwise it will attempt to drive through the basket, colliding on it.

Special conditions are present in some of the tasks, e.g. a termination signal for the cup falling off the table in the 'slide cup obstacles', as that would make the task impossible to solve for the remainder of the episode.

**Environment details.** The episode duration is fixed to a maximum of 200 steps. Invalid actions are rewarded with 0 and the agent is kept still, to discourage the agent from taking them. The maximum delta per action in EE, E3 and joint spaces has been fairly chosen to provide similar freedom of movement in task space to all approaches. Please refer to the accompanying code for further details.

## A.4  ADDITIONAL RESULTS

All experiments in the paper are run with the following amount of seeds:

- 5 seeds for E3J (ours), EE, Joint
- 3 seeds for E3A
- 2 seeds for E3J-WRIST (ablation)

Find the detailed results over training for the 8 RLBench tasks in Figure 11.

Find the detailed results over training for the ablation study on the full-body tasks in Figure 12.

**E3J controlling different joints.** In order to empirically verify the motivation for choosing the base joint (the first joint) as the joint that is controlled directly for E3J, we perform an ablation study on different variations of E3J, where we perform control by directly controlling different joints. The results are presented in Figure 13.

The first observation is that constraining the joints numbered with even numbers completely harms the agent's performance. This is expected, as the robot is equipped with 4 joints (oddly numbered) that rotate horizontally and 3 joints (evenly numbered) that rotate vertically, thus the redundancy lies among the oddly numbered joints and constraining one of the vertical ones can impair the robot's control. Overall, the results confirm the first joint to empirically be the most versatile choice. However, in the 'reach elbow pose' task, some joints actually allow faster learning than the first joint, demonstrating that using the base joint is an arbitrary choice and that there may be other tasks where direct control of a different joint would be more beneficial.

**E3J on different robots.** In order to empirically verify the general applicability of E3J to different robot models, we ran experiments on two additional robotic platforms: a Kuka iiwa 7 with Robotiq gripper and an xArm with xArm gripper (see Figure 14). The results are reported in Figure 15 on

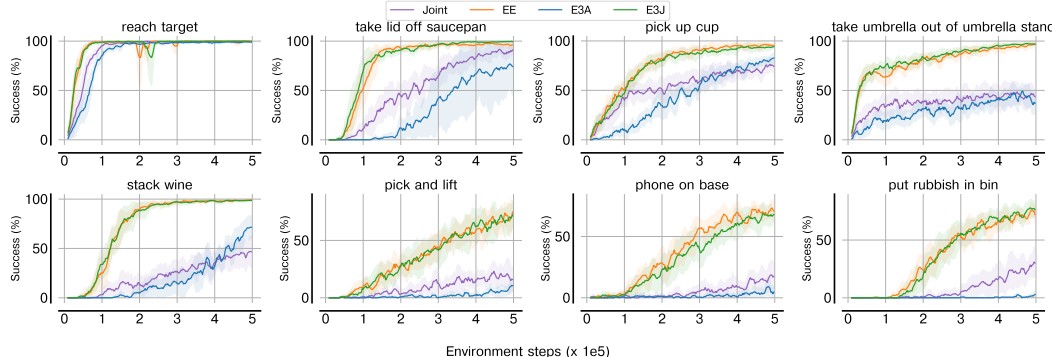

Figure 11: **RLBench detailed results.** Performance overt time on 8 RLBench tasks. Note, the goal here is **not to outperform EE on these tasks**, but to instead show that there is **no loss in performance** by using E3.

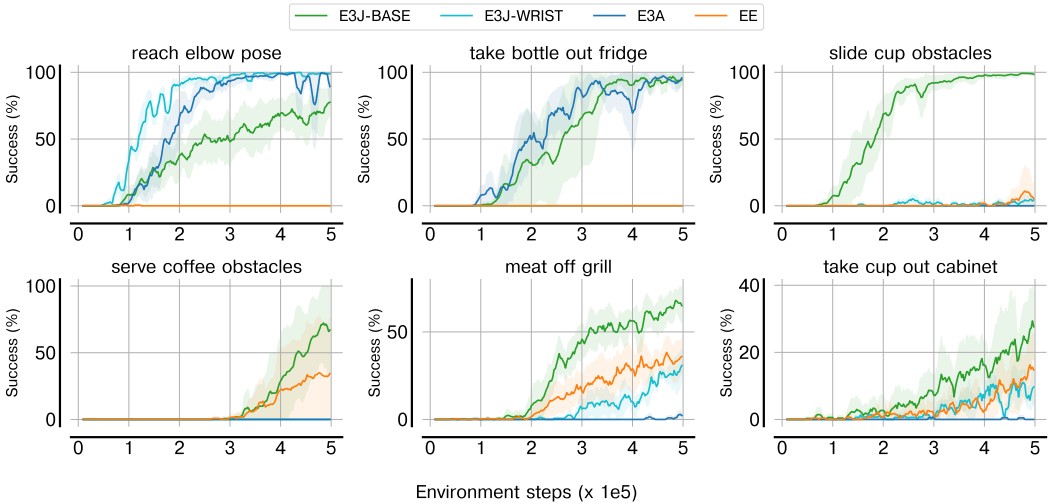

Figure 12: **Ablation detailed results.** Results over time on the 6 full-body tasks.

a set of 4 tasks, including 3 vanilla tasks from RLBench task and 1 full-body task (reach elbow pose). The results mostly reflect the same properties observed on the Panda platform: excluding the 'reach target' task, where all action spaces perform comparably, the joint space enables solving full-body tasks, such as 'reach elbow pose', but tends to learn slower in the other tasks; the EE space learns faster than joint in common RLBench tasks, but struggles in fully-body tasks. E3J inherits the advantages of both action spaces and also tends to learn faster in the 'take lid off saucepan' and 'stack wine' tasks. We also note that the learning curves look different from the Panda learning curves (see Figures 4 and 11), as the different morphology and controllability of the arm and gripper affect the performance of the agent.

**E3J with more than 7 joints.** Given the lack of availability of robot models with more than 7 joints to control, in order to empirically verify the applicability of E3J to robot arms with more than one redundant joint, we designed a custom 8 DoFs Panda robotic platform. The Panda has been modified to have two additional joints at the base of the robot, following the same structure of the original first two joints of the robot. The result is a Panda with 9 joints (see Figure 14), rather than 7, of which we chose to control 8 joints, thus excluding the base one. In this setup, there are two redundant joints to control. For the joint action space, the setup requires controlling 8 joints rather than 7. For the EE space, the setup requires adapting the IK computation to work with 8 joints rather than 7. For the E3J space, as we advised in the main text, we control two joints (the first two controllable joints

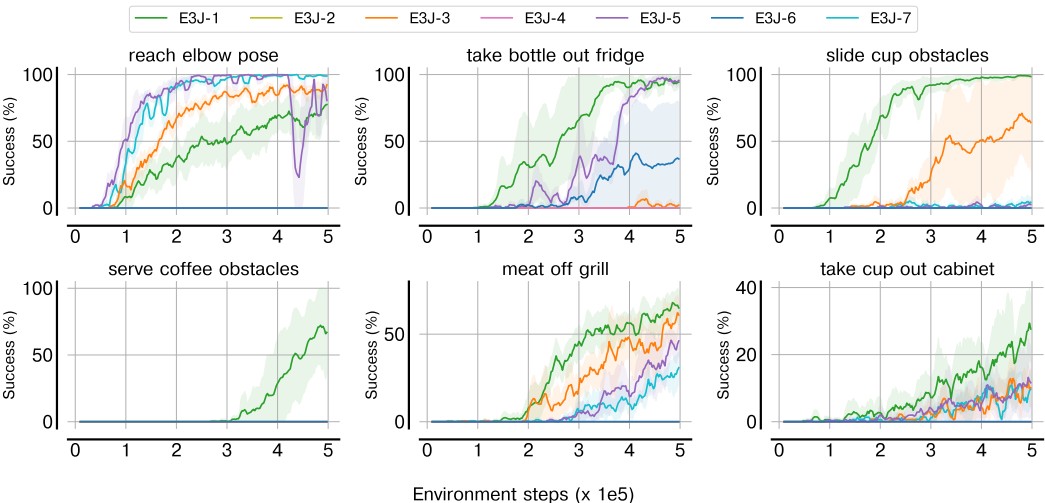

Figure 13: **E3J controlling different joints.** Results over time on the 6 full-body tasks, selecting different joints to control with the E3J method. (3 seeds)

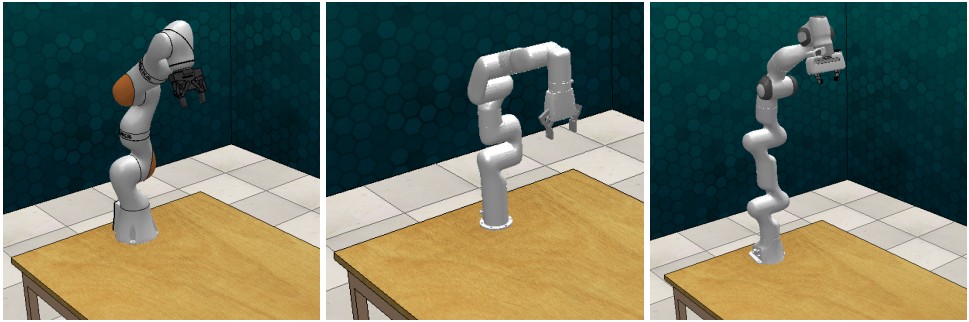

Figure 14: Additional robotic platforms to evaluate our approach. From left to right: KUKA LBR iiwa 7 with Robotiq gripper, xArm, and a custom Panda with 9 joints.

from the base) and, as done for 7 DoFs robots, compute the IK on the remaining six joints (with no redundant joints).

From the experimental results, presented in Figure 16, we observe that the main effects observed on the 7 DoFs Panda hold in this setup, with two main differences: (i) all methods take longer to learn, due to the increased action dimensionality and more complex morphology of the robot, (ii) E3J tends to be more efficient than EE on the 'take lid off saucepan' and 'stack wine' tasks, differently from the original Panda setup.

## A.5 SIMULATION EXPERIMENTS - SETUP

**SAC model and hyperparameters.** The agent receives as inputs two RGB images, from a third-view and a wrist cameras, and a set of proprioceptive states. RGB inputs have dimensionality 84x84x3 and are processed using a convolutional encoder, where random shift augmentations are applied during training (Yarats et al., 2020). No frame stack or action repeat is used. We did not account for contact forces, but the agent can infer contacts, by visual or proprioceptive feedback (e.g. the end-effector is blocked) or through the rewards (e.g. the agent is rewarded to stay in contact with some object to complete the task).

In order to increase stability we adopt the following common RL modifications to the original SAC: a distributional critic (Bellemare et al., 2017), reward and return scaling (Schaul et al., 2021). The learning rate is $3 \cdot 10^{-4}$. The agent learns by "stepping" 8 environments in parallel and is updated

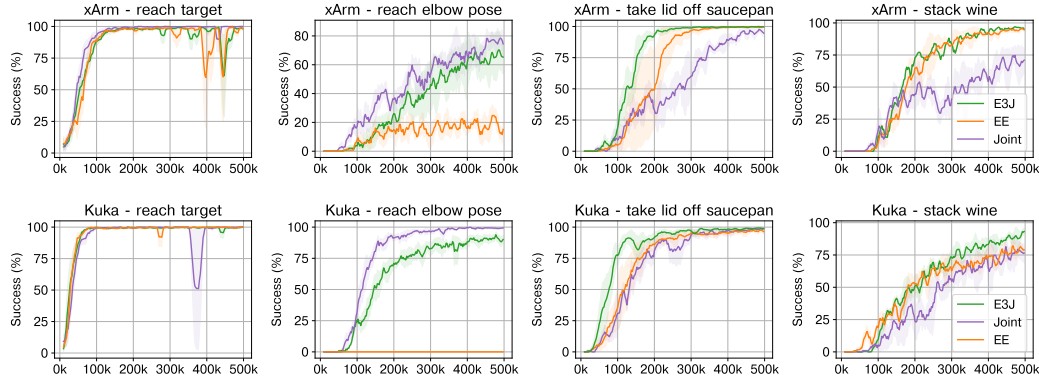

Figure 15: **E3J on different robots.** Results over time for 4 tasks on different robotic platforms: Kuka iiwa 7 and xArm. (2+ seeds)

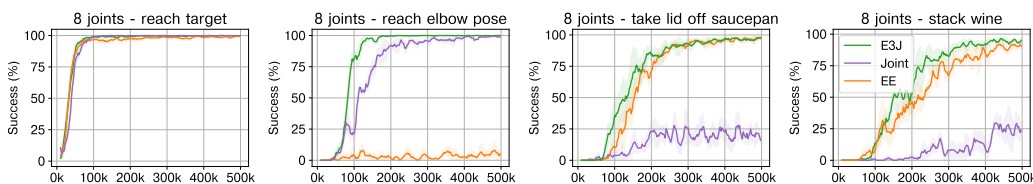

Figure 16: **E3J with more than 7 joints.** Results over time for 4 tasks on a custom Panda (with 9 joints) where the agent controls 8 joints. (2+ seeds)

after every step (1 update every step = every 8 environment steps). Please, refer to the accompanying code for further details.

**IK implementation.** We rely on the internal IK solver of CoppeliaSim for EE, E3J and E3A, which, for quickly computing the IK in delta action spaces, employs the pseudoinverse Jacobian method. For E3J, we fix the joint controlled separately to the desired value and compute the IK on the remaining joints, thus, solving the IK for 6 constraints that are dictated by the task space coordinates (position and orientation of the end effector). For E3A, we place an additional constraint on the arm angle, solving the IK with 7 constraints. Overall, IK-based methods (EE and E3) require around 35% more time to be executed in simulation, compared to direct control of the joints.

**E3J implementation choices.** For E3J, we know a change in the position of the joint that is controlled separately will affect the rest of the configuration. In practice, we can exploit this fact to simplify the understanding of actions for the agent. If we apply the transformation implied by the directly controlled joint variation to the actions of the agent, we let the agent work in a transformed task space, that is rotation-translated with respect to the world space. Consequently, understanding action validity and solving the IK problem becomes as easy as in the end-effector action mode, with the additional simplification that the IK solves a 6 DoF problem with no redundancies. Furthermore, the action space acquires useful features, such as rotation and translation invariance, after fixing the controlled joint. This strategy can be particularly useful, especially when controlling the base joint, as the agent can easily learn to perform variations of a task around itself, by correctly placing the base joint and providing the same actions for the other action space dimensions.

We employed E3J with rotated space around the shoulder rotational joint (the first "base" joint) for the main results presented in Section 5.

### A.6 REAL WORLD EXPERIMENTS - SETUP

**Hardware.** Franka Emika Panda with Franka Hand and one Intel® RealSense™ Depth Camera D455 attached to the wrist.

**IK implementation.** We use MoveIt and PickNikRobitics' pick_ik (`https://github.com/PickNikRobotics/pick_ik`) with default parameters for solving the IK for EE and E3J. We followed the standard "pick_ik Kinematics Solver" tutorial from the pick_ik documentation, which references the parameters from the "pick_ik_parameters.yaml" file. We only altered two parameters for the IK: changing both `approximate_solution_position_threshold` and `approximate_solution_orientation_threshold` from 0.05 to 0.01, to reduce the tolerance on the EE pose and thus increase accuracy.

pick_ik uses a local optimizer, which solves inverse kinematics via gradient descent, and a global optimizer, based on evolutionary algorithms. For E3J we first simulate the motion created by the joint that is controlled separately and then compute the IK on the remaining joints.

**ACT hyperparamters.** Each action space model was trained for 3000 epochs using the hyperparameters presented in the ACT paper (apart from using a batch size of 16, and chunk size of 20). Additionally, the EE pose was added to the state information ($\mathcal{S}_{ee} = (\mathcal{S}_T, \mathcal{S}_R)$, where $\mathcal{S}_T \in \mathbb{R}^3$ is EE position and $\mathcal{S}_R \in \mathbb{R}^4$ is EE rotation as a quaternion). All quaternions in demonstrations and inference were forced to have a positive $w$. Lastly, only a wrist camera (with resolution $224 \times 224$) was used.

## A.7 LIMITATIONS

Despite its advantages, singularity points still occur with E3J, as they are rarer than with EE but still present. E3A does not present such problems but has been shown to perform worse. Further investigation into strategies for mitigating singularities within E3J or enhancing the efficiency of E3A would prove valuable. Moreover, we have only investigated two realisations of the E3 action space (E3J and E3A); future work could look to investigate alternative means of achieving E3.

