# OpenReview forum: "End-Effector-Elbow: A New Action Space for Robot Learning"
_ICLR.cc/2024/Conference — Submitted to ICLR 2024_

### Official Review · Reviewer_Sd65 · 2023-10-31

**Soundness:** 2 fair
**Presentation:** 3 good
**Contribution:** 2 fair
**Rating:** 6
**Confidence:** 4

**Summary:**

This paper presents E3, a new action space for robot arm manipulation. Essentially, the proposed method presents an action space containing both end-effector pose and control over one additional joint in the arm. Compared to previous 2 established actions space, i.e. end-effector pose or joint-space, E3 brings the advantages of both, and show performance gain on a set of experiments.

**Strengths:**

- The idea is straightforward, reasonable and easy to follow
- the paper presentation is clear
- empirical results showing value of the proposed method

**Weaknesses:**

- My biggest concern is, the proposed method is more like a small engineering trick, rather than a rigorous scientific approach. It is specific to 7-DoF arm, which presents opportunities for such method due to the one additional redundant joint. The method won't work for 6-DoF arms, which although doesn't come with redundancy, but will also present multiple disconnected IK solutions given one end-effector pose, and only a subset of them would satisfy the task constraints.It also won't work for arms with DoFs greater than 7: which joint should we choose then? Or do we need to control 2 joints?
- Some explanations are not clear:
   - It says `elbow`, but in the method it says empirically it uses the base joint
   - i don't understand why choosing any joint in between would make problems for IK solver: it's just removing 1 DoF and the solver should work just fine. There's also no experiments to empirically validate which joint to choose

**Questions:**

See above

---

> ### Author Response · Authors · 2023-11-14
>
> We would like to thank the reviewer for their feedback. We provide an answer to their questions below.
>
> ***
>
> **The work can be seen as an engineering trick**: Contrary to viewing the 7 DoF as an inherent opportunity for E3, our new action space is a simple strategic response to this overactuation problem, commonly encountered in robot arms with greater than 6 DoF. The E3 action space applies classical robotics control techniques to Robot Learning (see "Redundant manipulators control" part of the "Related Work" section of the original paper), aiming to replace Joint and EE action spaces currently used in the robot learning literature.
>
> While the E3 action space offers a straightforward solution, its simplicity is its strength: it can be seamlessly applied to many existing and future robot learning works. Additionally, it is clear from our findings that this simple change in action space for robot learning dramatically improves performance on tasks where whole-arm control is imperative. As we confront increasingly complex challenges, such as navigating real-world scenes involving cupboards and other intricate, hard-to-navigate areas, it becomes evident that conventional action spaces fall short in providing the necessary control over the arm's configuration whilst maintaining data-efficient learning.
>
> Previous contributions, which also mostly focus on re-engineering the action space, e.g. by adopting primitives [1] or next-best pose [2], has had a significant impact on the field, leading to higher performance, through only minor modifications of the learning algorithm.
>
>
>
> **Generalization to $\neq$ 7 DoF arms**: In the case of a 6 DoF arm, such as the UR robot family, where the "elbow" has two solutions for end-effector (EE) poses (elbow up or elbow down), E3J method would utilize pose information along with a binary flag indicating elbow orientation. This contrasts with the conventional approach in robot control for UR robots, where the elbow joint is typically constrained to either the up or down position.
>
> Additionally, 6 DoF arms face limitations due to their morphology. Unlike overactuated arms, they may struggle with tasks demanding flexibility. For instance, consider our real-world setup with a cabinet except replacing the Franka Emika Panda arm with a UR5. The UR robot's vertical elbow position would hinder solving the task from this position where the elbow needs to be rotated to the horizontal place in order to reach inside the cupboard. Visual example in simulation available at https://imgur.com/a/n4L5UZH : while the cup position would be reachable by the UR5, the presence of the obstacle and the stiffness of the UR5's elbow do prevent the robot from reaching inside the cabinet.
>
>
> **Why the method talks about the elbow?** in most 7 DoF arms, the redundancy of the robot allows control over one more dimension, which can be identified in the position of the elbow. As we mention in the Introduction, E3A allows direct control of the elbow by controlling its angle, while E3J allows indirect control of the elbow by constraining one of the joints to fix the elbow position. Thus, both the methods presented allow control over the elbow redundancy (see Figures 2 and 3 from the original submission for further illustration).
>
> **Experiments to validate the choice of the joint**: in Section 4.4, we motivated our choice for mainly considering E3J-base and E3J-wrist in the first iteration of the manuscript. In order to further validate our choice, we have now completed the ablation study, where we attempt to constrain other joints of the robot for E3J (see Section A4 and Figure 13 in Appendix). The results show that our intuitions generally proved correct, but there are tasks where constraining other joints than the base or the wrist improves sample efficiency. We also believe E3A may constitute a more general strategy than E3J, but in practice we found that the agent struggles to directly control the elbow angle more than indirectly controlling the elbow through one constrained joint. As finding out what's the best action space for robot learning, in terms of full-body control and efficiency, is one of the main goals of this work, we have chosen E3J as the idea that best fits this definition according to empirical evidence.
>
> ***
>
> We hope our revision answers the reviewers' doubts and we look forward to any further suggestions to improve our work.
>
>
> [1] Accelerating Robotic Reinforcement Learning via Parameterized Action Primitives, *Dalal et al*
>
> [2] Q-attention: Enabling Efficient Learning for Vision-based Robotic Manipulation *James et al*

---

> > ### Author Response · Authors · 2023-11-17
> >
> > As the rebuttal period draws to a close, we would like to thank you again for your useful feedback on our work.
> > We hope that our clarifications, together with the additions to the revised manuscript and the new experiments we are including, have addressed your concerns.
> > Assuming this is the case, we would like to ask if you would be willing to update your review score. Otherwise, please let us know if you have any further questions.

---

> > > ### Author Response · Authors · 2023-11-21
> > >
> > > Hi Reviewer Sd65,
> > > I hope this message finds you well. We've thoroughly addressed your valuable feedback on our submission in our rebuttal.
> > > Could I kindly request a moment of your time to review our rebuttal? Thank you!

---

> ### Comment · Reviewer_Sd65 · 2023-11-22
> **Reviewer response**
>
> My concerns have been partially addressed, but this work's technical novelty is till not fully convincing to me. I will raise my score to a positive one.

---

> ### Author Response · Authors · 2023-11-22
>
> Dear reviewer,
>
> we would like to thank you for answering to our rebuttal.
>
> We are glad to have addressed some of your concerns. In addition, we would like to report that the following experimental studies have been added to the revised manuscript, to corroborate the general applicability of our method:
> *  **evaluation on different robotic platforms**: using the Kuka iiwa 7 and xArm. In this setup, we showed that E3J can be applied in other platforms than the Panda
>  * **evaluation on a robot with more than 7 joints**: given the absence of common 8 DOFs, we designed a custom Panda, with additional joints, where the agent can control 8 joints. In this setup, we showed that E3J can be applied to robotic arms with more than 1 redundant joint to control.
>
> The results can be found in the Appendix of the revised manuscript.
>
> We hope the additional evaluations address the concerns that have been raised about evaluating only on a single 7 DOFs arm and extending the work to arms with higher DoFs.

---

### Official Review · Reviewer_H84E · 2023-11-01

**Soundness:** 2 fair
**Presentation:** 2 fair
**Contribution:** 2 fair
**Rating:** 3
**Confidence:** 5

**Summary:**

This paper proposes a new action space for robot learning, dubbed End-Effector-Elbow. The basic idea for this new action space is to avoid the joint space redundancies that end-effector based control incurs (when N (degress-of-freedom) of a robot > 6).

To avoid these redundancies the paper extends the robot’s action space to 6 (EE) + (N - 6), where the latter part is controlling the redundant part of the joint space. In many robots (e.g. Franka-Pandas) these redundant dofs arise in the elbow of the robot and probably thus the name. The paper shows that their proposed action space works better in some RLBench tasks (although 4 of the tasks seem to be new tasks) and in some small-scale real-world experiments.

**Strengths:**

The overall motivation of designing an action space that enables fast learning while not being adversely affected by the null-space is an important problem statement. The overall paper is well written and seems to present an interesting solution to this problem. The experiments (although limited) do seem to suggest that the approach works.

**Weaknesses:**

I think the overall approach is highly engineered for particular scenarios and robots and is not general. For instance, in the E3J method (the main method), the paper proposes to constraint the base joint but that is an arbitrary choice. Certain parts of the task may find it beneficial to constraint the wrist joint. Further, there can be tasks, which use configurations where constraining the middle joint is useful. For instance, consider scenario where the franka arm has been rotated such that it lies flat and thus the middle joint acts as an elbow. Clearly the proposed approach is infeasible in these scenarios.

**Relation to existing work:** To some extent the E3J idea can be viewed from the lens of a hierarchical policy (where first a constrained joint is learned and then the EE-action). Similar idea of hierarchical control and learning in hierarchical action spaces especially with respect to null-spaces has been considered in prior work (Sharma et al.). Unfortunately, the paper is not cited (or discussed).

Infact, the idea of hierarchical composition in Sharma etal is more general since it composes arbitrary number controllers in null-spaces. This automatically leads to the E3J approach where first a joint is selected and then the IK is used under the constraint of this joint to reach the desired agent (thus automatically acting in the null-space). One difference, between Sharma et al. and the current work is that the former only considers task-space constraints and use task-space impedance control to control the robot. While the current work uses a joint-constraint and task-constraint/goal and use IK for control. However, there are prior works (as referenced in Sharma eta al), which do controller composition directly in the joint space.

Infact, it may actually be useful to combine this paper with Sharma etal. and consider more general hierarchical policies.  Finally, there is other related works which focus on learning constraints from demonstrations that should also be potentially cited.

Sharma et al. *Learning to Compose Hierarchical Object-Centric Controllers for Robotic Manipulation*

Lin et al. *Learning Null Space Projections*

Howard et al. *A novel method for learning policies from variable constraint data*

**Generality of the approach:** Can you talk about the challenges in using this approach on very differently settings. For instance, consider the UR arms and if the task only requires one degree of freedom for successful execution. The presented approach wouldn’t consider this scenario but wouldn’t the benefits of a constrained action space for learning be useful even in this setting?

It would also be useful to show images/videos of simulation tasks which are being considered as full-body tasks.

**Use of IK:** As is briefly noted in the paper, can the authors clarify if they indeed used constrained IK (with the joint constraint) to solve for the end-effector target pose? Were there any issues or challenges related with solving this optimization problem. For instance, what if the IK failed because the joint constraint makes it hard for the robot to reach the desired EE pose. I would imagine such scenarios to arise, but the paper is very light on such implementation details.

Another issue with an IK based approach is that it can be expensive to solve at every RL step especially when we perform delta end-effector actions. In such settings most works use the jacobian based controller (osc), how would the proposed approach work with such a choice?

**Questions:**

Please see above.

---

> ### Author Response · Authors · 2023-11-14
> **Rebuttal pt. 1**
>
> We would like to thank the reviewer for their feedback. We provide an answer to their questions below.
>
> ***
>
> **The approach is not general**: as pointed out by the reviewer, the idea of constraining one specific joint for E3J is an arbitrary choice and may not always be the most effective choice. In the original submission, we discussed this in Section 4.4, in the Joint Selection and Practical Implementation Choices paragraphs, where we motivated our choice for mainly considering E3J-base and E3J-wrist. In order to further validate our choice, we have now completed the ablation study, where we attempt to constrain other joints of the robot for E3J (find results in the updated manuscript's Appendix). The results show that our intuitions generally proved correct, but there are tasks where constraining other joints than the base or the wrist improves sample efficiency. We also believe E3A could be a more general strategy than E3J, but in practice we found that the agent struggles to directly control the elbow angle more than indirectly controlling the elbow through one constrained joint. As studying what's the best action space for robot learning, in terms of full-body control and efficiency, is one of the main goals of this work, we have selected E3J as the idea that best fits this definition according to empirical evidence.
>
>
> **Relation to existing work**: we found the work of Sharma et al, which presents a hierarchical framework to combine task-space constraints, to be relevant to our method, and thus we added a discussion about it in the revised manuscript. Overall, as pointed out by the reviewer, the work presents several differences from our approach, in the settings and the goal pursued: (i) it presents a hierarchical method for selecting actions, while our method focuses on learning flat policies; (ii) it applies task-space constraints, while we combine task-space control with joint constraints, (iii) the goal of Sharma et al's work is to present a modular framework that improves sample efficiency and generalization, while we explicitly focus on studying the most convenient action spaces that enable reinforcement learning or imitation learning methods to solve full-body tasks, whilst retaining the benefits of controlling the agent in task space coordinates. We discuss the work in the Related Work section of the revised manuscript (highlighted in green), along with citing the relevant literature about null-space projections and task-space constraints that have been recommended.
>
> Furthermore, the reviewer made a comment that an interesting direction could be to combine our constraints for controlling the elbow (controlling the elbow angle or one of the robot's joints) with a hierarchical controller (as in Sharma et al). This indeed sounds promising for future research, and so we added it in the Conclusion (see the revised manuscript, in green text).
>
>
> **Benefits of applying the approach in tasks that require less DoFs**: in order to compare the learning efficiency of our approach compared to EE on tasks that do not require full-body control, we presented results on 8 RLBench popular tasks, providing aggregated performance in the main paper and detailed results in the Appendix. We did not find any significant difference in performance between EE and E3 in these settings. However, we would like to highlight that 2 out of the 6 full-body tasks are also standard RLBench tasks (`meat off grill` and `take cup out cabinet`), requiring in theory <= 6 DoFs. However, the presence of obstacles in reaching the task's main object ends up favouring our method in terms of learning efficiency, as E3 provides the control necessary for consistent obstacle avoidance.
>
> **Images/videos of full-body tasks**: we added the requested visualizations, with a Figure in the Appendix, and with videos on the [project website](https://doubleblind-repos.github.io/).
>
> [continues]

---

> > ### Author Response · Authors · 2023-11-14
> > **Rebuttal pt. 2**
> >
> > [continues]
> >
> > **Use of IK:** we provided additional details in the Appendix on which IK solver we employed in simulation and real-world tasks (see updated text in green). We thank the reviewer for the suggestion of comparing the running time of different action spaces. For the delta action space experiments, similarly to OSC controllers, we employ a Jacobian-based solver for computing the IK, which takes around 35% more time than directly providing joint coordinates to the simulator. This important detail has also been added to the manuscript (see new green text in Appendix A.5).
> >
> > As we use a Jacobian-based solver the main issue with the IK failing comes from the inability of satisfying the delta coordinates provided by the agent as a target at each step. As we show in Figure 6b, the action's invalid rate is similar from the start for EE and E3J using the base joint, as opposed to E3A and E3J using the wrist joint. This led us to the hypothesis that the agent failing to provide IK-valid actions more consistently for E3J-WRIST and E3A might be the cause of underperformance, compared to E3J-BASE (Figure 6a).
> >
> > As OSC controllers generally employ Jacobian-based methods, similar issues might arise if we switched to these controllers for our action space. Nonetheless, it would be worth verifying such a hypothesis in future work, i.e. studying the effects of applying E3 under different underlying controllers.
> >
> > ***
> >
> > We hope the revision satisfies the reviewer's requests and we look forward to any further suggestions to improve our work.

---

> > > ### Author Response · Authors · 2023-11-17
> > >
> > > As the rebuttal period draws to a close, we would like to thank you again for your useful feedback on our work.
> > > We hope that our clarifications, together with the additions to the revised manuscript and the new experiments we are including, have addressed your concerns.
> > > Assuming this is the case, we would like to ask if you would be willing to update your review score. Otherwise, please let us know if you have any further questions.

---

> > > > ### Author Response · Authors · 2023-11-21
> > > >
> > > > Hi Reviewer H84E,
> > > > I hope this message finds you well. We've thoroughly addressed your valuable feedback on our submission in our rebuttal.
> > > > Could I kindly request a moment of your time to review our rebuttal? Thank you!

---

> ### Author Response · Authors · 2023-11-22
>
> Dear reviewer,
>
> we would like to report that the following experimental studies have been added to the revised manuscript:
> * **evaluation on different robotic platforms**: using the Kuka iiwa 7 and xArm. In this setup, we showed that E3J can be applied in other platforms than the Panda
> * **evaluation on a robot with more than 7 joints**: given the absence of common 8 DOFs, we designed a custom Panda, with additional joints, where the agent can control 8 joints. In this setup, we showed that E3J can be applied to robotic arms with more than 1 redundant joint to control.
>
> The results can be found in the Appendix of the revised manuscript.
>
> We hope the additional evaluations address some of the concerns that have been raised. In particular, the studies should show the more general applicability of our approach. Furthermore, regarding your concerns about the benefits of using a constrained action space in tasks that do not require full-body control, the new results show that our approach (E3J) tends to learn faster than EE, even in standard RLBench tasks, which only require grasping and moving objects to a target (such as `take lid off saucepan` and `stack wine`).

---

> > ### Comment · Reviewer_H84E · 2023-11-23
> > **Thank you for the response.**
> >
> > Thank you for your comments. I think this is an interesting submission but I think much more work needs to be done to make it a solid contribution. I would encourage the authors to further consider reviewer comments to improve the submission.
> >
> > > where we attempt to constrain other joints of the robot for E3J (find results in the updated manuscript's Appendix).
> >
> > The issue is not merely about which joint to constraint but more about the same task may require different joints to be constrained during different sub-tasks within the same task e.g. grasping a block and then putting it into a cabinet (side insertion) will require different joints to be constrained. I don’t think the proposed method can handle these scenarios and that is why the lack of generality.
> >
> > > we provided additional details in the Appendix
> >
> > I checked the details in the Appendix and they are really light on implementation. There are many oddities in the text as well. For instance, the paper states “solving the IK for 6 constraints.”, I am not really sure what to make of this statement? There are 6 variables in the objective function or 6 constraints in the optimization’s objective? There are multiple ways to solve for IK, which ones did the authors use? What happens when there exists no IK solution? All of these are important details that should be provided in the paper.
> >
> > > Related work
> >
> > What this paper seems to suggest is very similar to the idea of hierarchical control, which in my opinion can go beyond redundant manipulators. I know the latter is discussed in this paper but I would encourage the authors to consider more general hierarchical control works as well.

---

> ### Author Response · Authors · 2023-11-23
>
> Dear Reviewer H84E,
>
> we would like to thank you for participating in the rebuttal and providing additional comments on our work.
>
> **Choice of joint for E3J and generality of the approach**
>
> We understand the idea of "constraining one joint" of the robot may sound less general. However, we believe this is more of a wording issue and how the method is perceived than an actual issue of the generality of the approach.
>
> When using E3J, the outputs of the agent are organized as
> (x, y, z, yaw, pitch, roll, joints_controlled_separately), which can also be read as (task space coordinates, joints_controlled_separately). This means the agent provides the same control of the standard EE action space, plus control over additional joints. Clearly, this **extends the control of the agent over a larger set of joint configurations**, as it implies the agent can achieve EE poses, by setting six joints using the IK, and where there would normally be many arm configurations which achieve an EE pose, the additional controlled joints enable the agent to choose one configuration to move to. We suggest looking at Figures 2 and 3 to gain additional visual insights into how our method works.
>
> We empirically verified that our method provides more extensive control in the ablation introduced with the revision (now under the name `E3J controlling different joints`), where we showed that controlling any of the redundant joints (which are the oddly numbered ones - 1,3,5,7) allows control over the full-body pose of the robot, as shown in the `reach elbow pose` task, which requires precise control over the robot's configuration to be solved. The main difference in performance between controlling different joints arises from how convenient directly controlling one joint is, compared to the others, and we found controlling the base shoulder joint as empirically being the most versatile option.
>
> The reason why we used the term "constrain" was due to the underlying implementation, which works by setting the controlled joint position directly and computing the remaining joints' position using the IK, with one less constraint given that the joint is directly controlled by the agent. To avoid any further confusion, **we have replaced the term "constraining" with the term "controlling"**, as from the agent's perspective, the joint is directly controlled (see text highlighted in orange in the revised manuscript).
>
> **Implementation details**
>
> We ensured that all the implementation details are present on the paper as follows:
>
> - "What are the 6 constraints?"
>
> We apologise for the ambiguity about which constraints are being solved by the IK. We further specified that these are the task space coordinates of the end effector as follows: "solving the IK for 6 constraints that are dictated by the task space coordinates (position and orientation of the end effector)".
>
> - There are multiple ways to solve for IK, which ones did the authors use?
>
> This implementation detail is already described in the paper.
>
> For the simulation experiments, we mentioned:
>
> > "We rely on the internal IK solver of CoppeliaSim for EE, E3J and E3A, which, for quickly computing the IK in delta action spaces, employs the pseudoinverse Jacobian method."
>
> For the real-world experiments, we mentioned:
>
> > "We use MoveIt and PickNikRobitics’ [pick ik](https://github.com/PickNikRobotics/pick_ik) with default parameters for solving the IK for EE and E3J. For E3J we first simulate the motion created by the joint that is controlled separately and then compute the IK on the remaining joints."
>
> We also added the following information (in orange text), to provide easier access to how Pick IK works (and how we used it) to the reader:
>
> > "We followed the standard ["pick\_ik Kinematics Solver" tutorial](https://moveit.picknik.ai/main/doc/how_to_guides/pick_ik/pick_ik_tutorial.html#pick-ik-kinematics-solver) from the pick\_ik documentation, which references the default parameters from the [pick\_ik\_parameters.yaml file](https://github.com/PickNikRobotics/pick_ik/blob/main/src/pick_ik_parameters.yaml). We only altered two parameters for the IK: changing both approximate\_solution\_position\_threshold and approximate\_solution\_orientation\_threshold from 0.05 to 0.01, to reduce the tolerance on the EE pose."
> >
> > "pick\_ik uses a local optimizer, which solves inverse kinematics via gradient descent, and a global optimizer, based on evolutionary algorithms"
>
> (continues...)

---

> ### Author Response · Authors · 2023-11-23
>
> (...continues)
>
>
> * What happens when there exists no IK solution?
>
> This implementation detail is already described in the paper. In the Environment details section, we mentioned:
>
> > "Invalid actions are rewarded with 0 and the agent is kept still, to discourage the agent from taking them."
>
> and invalid actions are described in the main text as:
>
> > "Note that invalid actions can be caused by the Jacobian-based IK solver failing due to an impossible configuration request. This could be due to: (i) joint configurations that are too far from the current one and thus unreachable, (ii) attempts to collide with objects, or (iii) configurations outside the joint limits."
>
> We also would like to highlight that we have shared our code, so that any implementation detail can be verified for the simulated experiments, and **we will open source the code shared in the supplementary material upon publication** to foster further research in this direction (thanks to the new two action spaces, the four new tasks and the three new robots we introduced).
>
> **Hierarchical control**
>
> While the idea of hierarchical control would definitely be interesting to explore for future work, and we discussed this in the paper, as the reviewer acknowledged, **the current work is about a new action space, involving no hierarchy**.  Hierarchical control requires the agent to reason about actions in different stages while our work requires the agent to output a hybrid set of coordinates, in task and joint spaces, in one flat stage. Hierarchical control mostly employs actions that are temporally-consistent and last for a longer time, while our work focuses on atomic actions that should be provided at high frequency for closed-loop control.
>
> For example, in the work mentioned by the reviewer [1], the agent should reason about which controllers should be combined, by selecting multiple object-axis controllers in a prioritized order, which are composed together using null-space projections. In contrast, in our work, there is no priority as the action's dimensions have all the same priority, and thus the policy is "flat".
>
> Another example is primitives [2], where the agent should reason, first, about which primitive to use, and then, on the parameters for applying the primitive to the environment. Our approach jointly controls the robot's configuration by providing targets in task and joint coordinates, rather than prioritizing any of them.
>
> Extending our work to flexibly choose which joint to directly control with E3J is described as a future work direction in the Conclusion, and we believe our work, including all the assets we are releasing (new E3A and E3J action spaces, new full-body tasks to evaluate, new robot models to experiment with), will provide **an useful starting point and a solid baseline for future research in this area**.
>
> [1] Learning to Compose Hierarchical Object-Centric Controllers for Robotic Manipulation, *Sharma et al*
>
> [2] Accelerating Robotic Reinforcement Learning via Parameterized Action Primitives, *Dalal et al*
>
> ---
>
> We are encouraged that the reviewer finds our submission interesting and we hope that the above clarifications along with the revisions to the manuscript address the reviewer's remaining comments.
>
> If you are more satisfied with our revision, we would kindly ask you to update your score to reflect this.

---

### Official Review · Reviewer_XGZj · 2023-11-02

**Soundness:** 4 excellent
**Presentation:** 4 excellent
**Contribution:** 3 good
**Rating:** 8
**Confidence:** 4

**Summary:**

- Proposes a new action space called End-Effector-Elbow (E3) for robot arm control that allows both efficient learning and full control over the arm configuration by utilizing the extra degree of freedom in a 7-DoF robot arm used for 6-DoF EE control. Introduces two realizations of E3: E3Angle (E3A) which controls the elbow angle directly, and E3Joint (E3J) which controls the elbow position indirectly by fixing a joint.
- Shows through RL experiments in simulation that E3, especially E3J, outperforms joint and end-effector control in tasks requiring precise arm configuration. Real-world imitation learning experiments all show that E3 succeeds in confined spaces while end-effector control fails.

**Strengths:**

- Addresses limitations of standard joint and end-effector control spaces.
- Achieves better sample efficiency than joint control and better arm control than end-effector control.
- E3J alignment with task space enables efficient learning like end-effector control.
- Experiments show benefits in simulation and real-world settings in both generic simulation benchmarks and also a hard real-world manipulation setup.

**Weaknesses:**

- The E3 action space does not consider the dynamics or contact forces of the robot and the environment, which may affect the performance and stability of the robot learning algorithms.
- The E3 action space is only tested on 7 DoF robot arms, and may not generaliz to other types of manipulators with different degrees of freedom or kinematics.

**Questions:**

- Only evaluated on a 7 DoF arm. The framework may not extend well to arms with higher DoFs and 6 DoFs. How does this method extend to higher DoFs? Is there is a generalized version of this framework?

---

> ### Author Response · Authors · 2023-11-14
>
> We would like to thank the reviewer for their positive feedback. We answer the remaining questions as follows.
>
> ***
>
> **The evaluation does not consider dynamics/contact forces**: for all the approaches we tested, we control the robot by setting target joint positions derived from the agent's output action space: either directly with joint space or through Inverse Kinematics (IK) for EE and E3 action spaces. In regard to the observations/inputs, the agent receives as inputs two RGB images, from the environment and wrist cameras, and a set of proprioceptive states. We did not account for contact forces, however in the RL experiments the agent can infer contacts, by visual or proprioceptive feedback (e.g. the end-effector is blocked) or through the rewards (e.g. the agent is rewarded to stay in contact with some object to complete the task). Nonetheless, we understand this would be useful to know for future readers and thus we added some additional comments on this, when discussing the agent's inputs from the environment, in the revised manuscript (see green text in Appendix A.5).
>
>  **Extension to higher DoFs**: arms with more than 7 DoFs are less common than 6 and 7 DoFs arms, and they are often designed for specific applications, which makes them hard to evaluate in common environments. We discussed this choice of focussing mostly on 7 DoF arms in Section 4.2 (highlighted in blue from the original submission) and, in order to provide a more general idea on how to extend our framework, we provided additional comments (in green) in Section 4.4, discussing how E3J could be adapted for higher DoFs.
>
> ***
>
> We hope our revision responds to the reviewer's questions about our work and we look forward to any further suggestions to improve our work further.

---

> > ### Author Response · Authors · 2023-11-17
> >
> > As the rebuttal period draws to a close, we would like to thank you again for your useful feedback on our work.
> > We hope that our clarifications, together with the additions to the revised manuscript and the new experiments we are including, have addressed your concerns and reinforced the positive opinion towards acceptance of our work. Please let us know if you have any further questions.

---

> > > ### Author Response · Authors · 2023-11-21
> > >
> > > Hi Reviewer  XGZj,
> > > I hope this message finds you well. We've thoroughly addressed your valuable feedback on our submission in our rebuttal.
> > > Could I kindly request a moment of your time to review our rebuttal? Thank you!

---

> > > > ### Comment · Reviewer_XGZj · 2023-11-21
> > > >
> > > > thanks for the rebuttal! I would like to invite the authors to address the concerns raised by other reviewers.

---

> > > > > ### Author Response · Authors · 2023-11-22
> > > > >
> > > > > Dear reviewer,
> > > > >
> > > > > we would like to thank you for answering to our rebuttal.
> > > > >
> > > > > In addition, we would like to report that the following experimental studies have been added to the revised manuscript, to corroborate the general applicability of our method:
> > > > > *  **evaluation on different robotic platforms**: using the Kuka iiwa 7 and xArm. In this setup, we showed that E3J can be applied in other platforms than the Panda
> > > > >  * **evaluation on a robot with more than 7 joints**: given the absence of common 8 DOFs, we designed a custom Panda, with additional joints, where the agent can control 8 joints. In this setup, we showed that E3J can be applied on robotic arms with more than 1 redundant joint to control.
> > > > > The results can be found in the Appendix of the revised manuscript.
> > > > >
> > > > > We hope the additional evaluations address the concerns that have been raised about evaluating only on a single 7 DOFs arm and extending the work to arms with higher DoFs.

---

### Official Review · Reviewer_AKTt · 2023-11-11

**Soundness:** 3 good
**Presentation:** 4 excellent
**Contribution:** 2 fair
**Rating:** 3
**Confidence:** 4

**Summary:**

It is common in robot learning to use either an end-effector or joint position action space. Many robots are 7-dof, where as EE action spaces are 6 dof, which means that there is an overactuated degree of freedom, leading to consistency issues. Most of the tasks and corresponding cost functions are metric, thus not aligned with joint action space. This paper proposed EE3 which overcomes both challenges by adding a joint angle (usually base or wrist to make sure there is kinematic consistency) to EE action space. The approach is evaluated on RLBench, where they show that this action space can boost performance in 6 RL tasks which require a good amount of obstacle avoidance, and performance does not drop in other tasks. The paper also shows results on 3 real-world tasks on the Franka arm.

**Strengths:**

- Rethinking action spaces and adding inductive biases for downstream tasks is an important problem
- EE3 is well motivated and the presentation of the problem statement is extremely clear
- The experiments are insightful, as we can see that EE3J helps in downstream robot tasks
- The real world experiments support the hypothesis

**Weaknesses:**

- I think there is a lack of discussion of other action spaces (primitives, OSC etc)
- There is a lack of comparison to other action spaces as well

- I don't fully believe this approach is novel - it can be seen in some ways as a special case of [1]. I believe the authors should discuss this further.
- It would be good to see this applied to other robots than Franka (including those with higher Dof).
- There are also concerns about how general this approach can be - it is only applicable to overactuated manipulation scenarios. It would be good to discuss this more in the paper.

[1] Riemannian Motion Policies. Nathan Ratliff, Jan Issac, Daniel Kappler, Stan Birchfield, Dieter Fox. 2018

**Questions:**

See weaknesses

---

> ### Author Response · Authors · 2023-11-14
>
> We would like to thank the reviewer for their feedback. Please find our responses to the questions below.
>
> ***
>
> **Lack of discussion/comparison to other action spaces**: the reviewer mentioned primitives and OSC as potential action spaces to compare with E3. We believe that, while both ideas are very relevant to the robot learning setting we work on, they are also orthogonal to our approach. Primitives, as in [1,2], provide a more abstract level for controlling the agent, through temporally consistent behaviors (often parameterized in task space coordinates). OSC controllers provide an alternative to IK solvers for controlling the robot in task space coordinates. In both cases, when working with a 7 DoF robot, additional control over the robot configuration needs to be provided to the learning agent for solving tasks in constrained spaces.
>
> In our work, we discuss and implement two ways (E3J and E3A) to address such limitations, and we test them in RL and IL settings, in a temporally fine-grained setting and using IK solvers. Nonetheless, we agree with the reviewer that testing our approach using OSC controllers, or defining primitives to control the remaining degrees of freedom of the agent, i.e. the elbow of the robot, are relevant ideas to mention for additional evaluations/future work and thus we discuss them in our revised manuscript (highlighted blue content in the Related Work section about primitives and OSC, new green content about OSC controller comparison in Section 5.1/Analysis).
>
> **Is the work a special case of Riemannian Motion Policies (RMP; Ratliff et al)?** We acknowledge that RMP represents one more way of controlling the elbow of a 7 DoF robot, and thus we added it to the "Redundant manipulators control" paragraph in the related work section, where we discuss other ideas to control the elbow. However, the nature of our work is very different from RMP: we aim to specifically discuss the issue of controlling the full-body pose for 7 DoFs robots, using data-driven learning techniques such as reinforcement and imitation learning. We discussed and empirically validated two implementations, and we acknowledge in the Discussion section that there are many opportunities to extend our study, along which we now also mention RMPs (see Related Work and Conclusion in the updated manuscript).
>
> **Testing on other robots than Franka**: we understand that an evaluation on multiple robot platforms could be more insightful and we have started experimentation with two other 7 DoFs robots: XArm 7 with XArm gripper and Kuka iiwa 7 with Robotiq gripper, and we aim to update the manuscript with partial results before the end of the rebuttal period. We will post additional comments when the results are ready.
>
> **Specify the method is designed for overactuated manipulation scenarios**: The initial submission made this statement in the Abstract, Introduction, Methods and Conclusion sections (see parts highlighted in blue in the manuscript). To further make it clear, we specified this additional times along the manuscript (in green, see the mentions to "overactuacted arms").
>
> ***
>
> We hope our revision responds to the reviewer's doubts about our work and we look forward to any further suggestions to improve our work further.
>
> [1] Accelerating Robotic Reinforcement Learning via Parameterized Action Primitives, *Dalal et al*
>
> [2] Augmenting Reinforcement Learning with Behavior Primitives for Diverse Manipulation Tasks, *Nasiriany et al*

---

> > ### Author Response · Authors · 2023-11-17
> >
> > As the rebuttal period draws to a close, we would like to thank you again for your useful feedback on our work.
> > We hope that our clarifications, together with the additions to the revised manuscript and the new experiments we are including, have addressed your concerns.
> > Assuming this is the case, we would like to ask if you would be willing to update your review score. Otherwise, please let us know if you have any further questions.

---

> > > ### Author Response · Authors · 2023-11-21
> > >
> > > Hi Reviewer AKTt,
> > > I hope this message finds you well. We've thoroughly addressed your valuable feedback on our submission in our rebuttal.
> > > Could I kindly request a moment of your time to review our rebuttal? Thank you!

---

> > ### Author Response · Authors · 2023-11-22
> >
> > Dear reviewer,
> >
> > we would like to report that the following additional experimental studies have been added to the revised manuscript, to corroborate the general applicability of our method:
> > *  **evaluation on different robotic platforms**: using the Kuka iiwa 7 and xArm. In this setup, we showed that E3J can be applied in other platforms than the Panda
> >  * **evaluation on a robot with more than 7 joints**: given the absence of common 8 DOFs, we designed a custom Panda, with additional joints, where the agent can control 8 joints. In this setup, we showed that E3J can be applied on robotic arms with more than 1 redundant joint to control.
> > The results can be found in the Appendix of the revised manuscript.
> >
> > We hope the additional evaluations address the concerns that have been raised about testing on other robots than the classic Franka Panda (including those with higher DOFs).

---

### Author Response · Authors · 2023-11-14
**General comment**

We would like to thank the reviewers for their time and insightful feedback. We are encouraged they found our problem statement to be important (H84E, AKTt) and thus our motivation to be clear, addressing the limitations of standard joint and end-effector action spaces (XGZj). We are pleased they found our experiments to be insightful (AKTt), showing the benefit and value of the new E3 action space (XGZj, H84E, Sd65) with our real-world experiments supporting our hypothesis (AKTt, XGZj). We are also glad the reviewers found our paper to be well written (H84E), clear (Sd65), and with excellent presentation (AKTt, XGZj).

We have responded to all the reviewers, under their comments, and updated the manuscript according to their suggestions (highlighted content from the original manuscript is in blue and new content is in green). We added an ablation study of E3J when constraining the different joints of the robot, which can be found in the Appendix of the manuscript, and we are also running additional experiments, with other robot models than Panda, to corroborate the general applicability of the method.

Some reviewers asked us to compare and relate our work with some approaches in the areas of hierarchical control, motion generation, or control strategies (e.g. OSC). While we acknowledge that all of these research areas are very relevant to the problem of controlling (overactuated) robot arms, we would like to emphasize that our work focuses on studying what's the best **action space** for robot learning, where with action space we intend the space that contains all the agent's learnable actions, i.e. the output of the learned policy network. The choice of the action space, defining which coordinates/constraints can be the outputs of the agent learned policy, is generally an orthogonal choice to how these coordinates are achieved on the robot (e.g. using an IK-based or OSC-based controller), or combined into higher-level motions/primitives for hierarchical control. In order to clear this confusion, we have further highlighted this definition in the Introduction and removed some phrasing that may cause confusion to the reader (e.g. joint control, in favour of joint actions).

We hope to have adequately addressed the reviewer's doubts so far and we remain available for any further comments.

---

### Author Response · Authors · 2023-11-22
**Summary of the rebuttal**

As the rebuttal period is ending, we would like to thank again all the reviewers for the useful feedback they provided. We replied to each reviewer's comments individually and we uploaded a revised version of the manuscript, hoping to address the concerns presented.
We summarize all the changes (highlighted in green and orange in the revised manuscript) as follows:
* **new visualizations** : we uploaded videos of E3 solving the full-body tasks we presented in our work on the [project website](https://doubleblind-repos.github.io/) and provided visualizations of the tasks in Appendix.
* **ablation on the choice of the joint for E3J**: we performed an ablation study of the effects of varying the joint controlled with E3J to deal with the redundancy in the manipulator. The ablation study can be found in the Appendix (Figure 13)
* **testing on more robotic platforms**: we evaluated our approach on other robotic platforms than the Panda, namely a KUKA iiwa 7 (with Robotiq gripper) and an XArm. The new experiments, showing the general applicability of our method, can be found in Appendix (Figure 15).
* **controlling more than 7 DOFs**: we added a discussion on how to extend the E3J method to robots with more than 7 DOFs, by constraining more than one joint (see E3J section). For evaluating our claim, given the absence of common manipulators with 8 DOFs, we designed a custom 8 DOFs Panda arm where we tested all the different action spaces. The study, confirming that E3J can scale to more than 7 DOFs, can be found in the Appendix (Figure 16).
* **relation to existing work**:  we acknowledged the works mentioned by the reviewers to be related to our work, despite not being directly comparable, due to the many differences in the experimental setup and the different nature of the problems addressed. For this reason, we have addressed this concern by adding descriptive comparisons, in the Related Work and Experiments sections of the manuscript, and describing integrations with such methods as potential future works, in the Conclusion.
* **clarifications about the problem**: in order to make clear the focus of our work, studying action spaces for (non-hierarchical) robot learning, in the context of overactuated arms, we added some clarifications in the Introduction and Method (E3) sections.
* **experimental details**: we included additional details about our experimental setup in the Experiments and Appendix sections.

If any further concerns arised during the final review of our work, that could not be signalled during the rebuttal period, we would be happy to address them in a potential camera-ready version of the paper.

---

### Meta-Review · Area_Chair_A9zb · 2023-12-12

**Metareview:**

Instead of vanilla cartesian control of end-effector pose or joint space control, this paper proposes a practical blend of the two for 7 DOF "overactuated" arms: control the base joint angle separately from 6-DOF end-effector pose (called E3J-base in the paper).  Once the base is set to a desired position, the robot is controlled in the task space by solving IK but now with one less degree of freedom. Variations of these ideas are proposed and demonstrated to be lead to faster learning on simulated RL benchmarks.

**Justification For Why Not Higher Score:**

In Sim-RL settings, the method is compared with standard alternatives and nothing much else - and in many of the plots the variance seems largish.  In Imitation Learning settings in the real world, vanilla joint control appears to perform just as well as the proposed approach.   Related work on unifying cartesian and joint space control shows much richer whole body behaviors, but is not compared with.  Overall, the paper lacks comprehensiveness when it comes to impact of action space design for control tasks, and novelty in relation to prior methods for fusion of motion generation methods, hierarchical control and whole body control.

**Justification For Why Not Lower Score:**

N/A

---

### Decision · Program_Chairs · 2024-01-16

Reject